# Mark-release-recapture of male *Aedes aegypti* (Diptera: Culicidae): Use of rhodamine B to estimate movement, mating and population parameters in preparation for an incompatible male program

**Brendan J. Trewin** [1]*, **Daniel E. Pagendam** [2], **Brian J. Johnson** [3,4], **Chris Paton** [3,4], **Nigel Snoad** [5], **Scott A. Ritchie** [3,4], **Kyran M. Staunton** [3,4], **Bradley J. White** [5], **Sara Mitchell** [5], **Nigel W. Beebe** [1,6]

**1** CSIRO Health and Biosecurity, Brisbane, Australia, **2** CSIRO Data61, Brisbane, Australia, **3** College of Public Health, Medical and Veterinary Sciences, James Cook University, Cairns, Australia, **4** Australian Institute of Tropical Health and Medicine, James Cook University, Cairns, Australia, **5** Verily Life Sciences, San Francisco, California, United States of America, **6** School of Biological Sciences, University of Queensland, Brisbane, Australia

* brendan.trewin@csiro.au

## Abstract

Rapid advances in biological and digital support systems are revolutionizing the population control of invasive disease vectors such as *Aedes aegypti*. Methods such as the sterile and incompatible insect techniques (SIT/IIT) rely on modified males to seek out and successfully mate with females, and in doing so outcompete the wild male population for mates. Currently, these interventions most frequently infer mating success through area-wide population surveillance and estimates of mating competitiveness are rare. Furthermore, little is known about male *Ae. aegypti* behaviour and biology in field settings. In preparation for a large, community scale IIT program, we undertook a series of mark- release-recapture experiments using rhodamine B to mark male *Ae. aegypti* sperm and measure mating interactions with females. We also developed a Spatial and Temporally Evolving Isotropic Kernel (STEIK) framework to assist researchers to estimate the movement of individuals through space and time. Results showed that ~40% of wild females captured daily were unmated, suggesting interventions will need to release males multiple times per week to be effective at suppressing *Ae. aegypti* populations. Males moved rapidly through the landscape, particularly when released during the night. Although males moved further than what is typically observed in females of the species, survival was considerably lower. These unique insights improve our understanding of mating interactions in wild *Ae. aegypti* populations and lay the foundation for robust suppression strategies in the future.

**Data Availability Statement:** Supplemental data is stored within the manuscript. Stored male recapture data and R code for the STEIK framework is available at https://github.com/dpagendam/MRRk; Raw mosquito and trap data is available (under embargo) at https://data.csiro.au/collections/collection/CIcsiro:47209v1#/collection/CIcsiro:47209.

**Funding:** This work was supported by the Australian National Health and Medical Research Council (NHMRC 1082127). NB received this award https://www.nhmrc.gov.au/. Project partners at Verily Life Sciences provided assistance in study design and reviewed the final manuscript but played no role in data collection, analysis or the decision to publish.

**Competing interests:** The authors have declared that no competing interests exist.

## Author summary

Incompatible insect techniques for controlling populations of the dengue vector, *Aedes aegypti*, utilize the mating biology of adult male mosquitoes to achieve suppression through a sterilization process. As the study of *Ae. aegypti* control has typically focused on adult female mosquitoes, knowledge on male movement, survival and mating interactions in the field is lacking. Here we undertook several mark-release-recapture experiments on adult male *Ae. aegypti* in Innisfail, Australia, and measured important biological parameters. For the first time in large field experiments, we employed rhodamine B as a marker that when fed to adult males, identified both marked males and the wild females they mated with. We observed males moving further through the landscape, but surviving for a shorter period, than previous measurements undertaken on females in a field setting. A high proportion (~40%) of unmated females suggests individuals are constantly available for mating. As such, sterile male strategies may need to release at regular intervals to achieve effective population suppression. The unique insights provided by this study will assist in designing future sterile male field interventions.

## Introduction

Rapid human population growth and urbanization, combined with widespread resistance to insecticides, have led to a dramatic increase in the incidence of vector-borne diseases such as dengue, chikungunya and Zika [1,2]. In the battle to contain widespread epidemics of vector-borne disease, mosquito control has turned to species-specific technologies to suppress mosquitoes and the pathogens they transmit at the landscape scale. Rapid advances in molecular biology, genetics and digital support systems have enabled area-wide 'rear-and-release' strategies such as the use of *Wolbachia* induced cytoplasmic incompatibility (or incompatible insect technique IIT) [3], the sterile insect technique (SIT) [4] and the *Wolbachia* population replacement method [5]. Together rear-and-release strategies are revolutionizing the suppression of mosquito-borne disease as they give rise to the 'fourth great era of vector control' [6].

For many decades mark-release-recapture (MRR) studies have been used to understand mosquito movement and population parameters [7]. Releasing marked individuals into a population allows for the inference of ecological parameters from both released insects and the wild population. Such studies provide estimates of mosquito movement, survival and population size via the Lincoln-Peterson Index (LPI) or its variations [8], all of which have been key to the management of disease spread in the past [9]. Traditional mosquito MRR studies have typically focused on adult female movement and ecology, because it is this population that drives pathogen transmission [10–12]. In contrast, the movement and mating behaviour of male mosquitoes is rarely a major component of MRRs, particularly in *Aedes aegypti* (Linnaeus), one of the world's most highly studied mosquito species [9].

Estimating male *Ae. aegypti* biological parameters, such as survival, dispersal and mating competitiveness, have become increasingly important as SIT/IIT methods rely on the success of mating interactions. Studies examining male *Ae. aegypti* movement have generally reported considerable variation. Early movement studies suggest the majority of male *Ae. aegypti* disperse within 50 m of release sites after one week [12–16], although these estimates may be attributable to small radi in trapping arrays [9]. Increasingly, studies have enlarged trapping distances and improved marking methods and statistical methods for estimating movement distances of both male and female *Ae. aegypti* [17–19]. Recent dispersal studies suggest that males move further than females, with mean distances travelled (MDT) of between 196 m and

294 m over experimental periods [17–19]. The average life expectancy (ALE) of male *Ae. aegypti* has been estimated to be between 1 and 3 days for both wild-type [13,20,21] and transgenic males [22,23]. The final parameter, mating competitiveness between modified and wild strains, is generally inferred from oviposition results in cage or semi-field cage trials [24–26] and rarely in a field setting [27]. All three of these biological parameters are essential to monitor the performance of mass-reared male *Ae. aegypti* as they affect the ability of males to efficiently seek out and successfully mate with wild-type females.

The primary challenge for SIT/IIT strategies is the determination of adequate release numbers to supplant matings with wild males (or "over-flooding ratio") across large areas. To do so, one must have a thorough understanding of target population size, demography, and movement within the landscape. Empirically informed models that simulate movement over extended landscapes are cost-effective methods of predicting release efficiency. Standard measures of population dynamics can often be obtained easily enough through traditional surveillance methods, but determination of movement patterns beyond that achieved in limited MRR studies is difficult. Traditionally, movement has been measured by summary metrics of flight such as a mean or range of the distance travelled, assuming movement is a discrete linear distance from the release point to traps [19,28,29]. More recently, mosquito movement studies have incorporated dispersal kernel theory, where distributions of movement can be estimated over the entire flight range [18,23] by integrating a temporal component such as average life expectancy [19]. Accurately parameterizing models for forecasting dispersal is a challenge, primarily due to the lack of accurate data and the expense of collecting these data from field environments. Furthermore, the development of precise models and simulations can only be achieved if field data from multiple ecological and environmental contexts are available to validate results.

A variety of marking methods have been employed to infer mosquito movement and behaviour including paints, dyes, trace elements, fluorescent dusts and radioactive and stable isotopes [7]. Marking methods are often limited in their effectiveness due to time inefficiencies in application, ability to detect markers, high expense, requirements for technical expertise and physical restrictions imposed by the mark on individual behaviours [7,30]. The fluorescent dye rhodamine B is a recent innovation in the use of fluorescent markers to stain male spermatophores in insects and has provided a rapid and cheap way to understand mating interactions [31–35]. Rhodamine B provides field ecologists with a method to measure both movement and mating interactions through the staining of male sperm, seminal fluids and body tissues. Producing a distinct bright red colour fluorescenece when excited under 540 nm (maximum excitation) and 568 nm (maximum emission) light wavelengths, the dye can be observed in ~95% of mated female *Ae. aegypti* spermathecae after four days [31]. The method allows investigators to mark both male and female mosquitoes, determine key performance indicators and rapidly infer the efficacy of an intervention by measuring behavioural and ecological factors such as mating success.

In preparation for a large, community-scale rear-and-release IIT program (Debug Innisfail, Australia, 2017–2018), we aimed to quantify the movement and mating behaviour of male *Ae. aegypti* through urban landscapes in north Queensland. To do this we undertook a number of rhodamine B-based MRR experiments, utilizing wild-type male *Ae. aegypti* to examine key biological parameters across a number of spatial, temporal and climatic scenarios.

## Methods

### Ethics statement

Human ethics was sought through the CSIRO Social and Interdisciplinary Science Human Research and Ethics Committee (CSSHREC) and approved under project 026/16 named

"Sterile insect technology development for *Aedes aegypti* ". As part of this approval all residents in release areas provided written consent for scientists to operate within their property, and were provided with an information sheet detailing how, why, where and when the research was to be performed and funding bodies. All residents were informed about the risks and benefits, including the potential for an increase in mosquito numbers during male releases. To enhance communication, brochures were distributed to homeowners, articles were posted in local newspapers, a website was setup for enquiries and residents were engaged through a project advisory group containing members of the local community.

## Study sites

Six mark-release-recapture experiments were performed during two seasonal periods, representing dry and wet seasons, in North Queensland, Australia. Mark-release-recapture experiments 1–3 (season 1) occurred late dry season, between 18 November and 13 December 2016, while MRR experiments 4–6 (season 2) occurred during the wet season, between the 7 and 27 February 2017. The study site in South Innisfail (17.5435°S, 146.0529°E) was situated in a residential area, 0.18 km² in size to the south east of Innisfail, a rural town on the main highway 88 km south of Cairns. The site contained 95 residential premises bounded by the Johnson River to the West and by grass sports fields and forest to the east. The site also contained a primary school to the north and a number of small commercial buildings (Fig 1). The Innisfail region is one of the wettest in Australia, averaging 3,547 mm of rainfall annually with tropical cyclones occurring throughout Summer and Autumn [36]. The urban landscape of Innisfail is unusual for northern Australia, with dwellings in the town a mix of Queenslander (constructed of wood with tin rooves and typically raised off the ground by 1.5–2 m) single floor fibre board, modern brick single floor, and 'art deco' style single floor constructions. House block size were approximately 800 m² with simple fencing or hedge-like greenery on boundaries, with open space underneath raised buildings utilized for storage, laundry and recreation areas. Roads averaged 25 m wide (fence to fence).

## Rearing and release

*Aedes aegypti* colonies were newly established with wild type eggs collected from multiple ovitraps in Innisfail before each experimental period. Mosquito colonies were maintained using

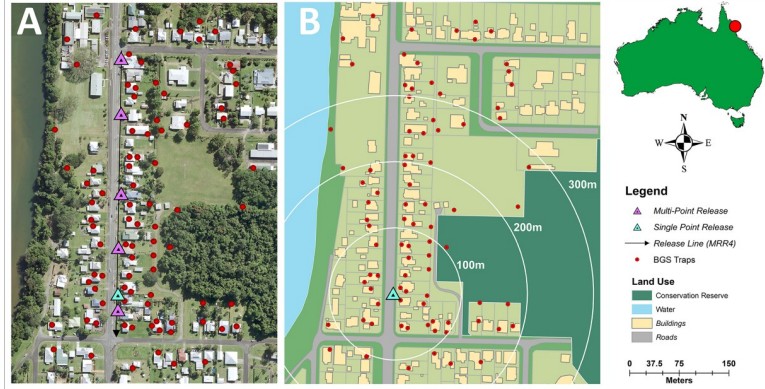

**Fig 1. Location of study site in South Innisfail, Australia.** Maps indicate landscape characteristics which include natural imagery (1A) and land use (1B). Rhodamine B marked *Aedes aegypti* were released at single point (blue triangle) and multi-point locations (purple triangles) and recaptured using Biogents Sentinel traps (red circles). Base layer imagery of South Innisfail (1A) provided by State of Queensland [2018] under licence [37] and landuse basemap (1B) digitized manually [38].

standard laboratory rearing protocols with 28˚C ± 1˚C, a 70% (± 10%) relative humidity, and a 12:12 hour light cycle and twilight period. F1 –F3 generation wild-type *Ae. aegypti* larvae were hatched into a solution of 0.2 g/L yeast in water, in which they were allowed to feed for 24 hours. Five-hundred first and second instar larvae were pipetted into a three-litre bucket to an approximate density of one larvae per 6 ml of water. Larvae in each bucket were fed ground Tetramin Tropical Fish Flakes (Tetra, Germany) provided at 0.45 g on day 2, 0.8 g on day 5 and again on day 6 if required. Ten minutes after the food settled, bucket water was stirred in a 'side to side' motion to distribute ground fish flakes. Male pupae were separated with a one ml bulb pipette based on size with 20 individuals placed into 300 ml Styrofoam rearing cups covered with mesh. After emergence, cups were visually inspected for the presence of females and if detected these were removed through aspiration. Adult males were fed a 0.4% rhodamine B (weight to volume) in a solution consisting of 160 mg rhodamine B dissolved in 40 ml of a 25% honey solution following the methods of Johnson et al. [31]. Males were maintained on the solution for four days to ensure adequate body and seminal fluid marking [31]. Males were transported to the study sites the day before release and released when five days of age.

Approximately 1,250 males were released during each of the six MRR experiments, with a delay of between seven and nine days between releases to separate recaptures. Releases occurred at 6am for day releases for MRR 1–5 and 7 pm for the night release (MRR 6). Release location varied depending on experimental design, with single point releases occurring at the southern end of the study site (MRR 1, 2 & 6; Fig 1B). For multi-point releases, males were divided evenly and released at five points along the eastern side of the central road (MRR 3 & 5; Fig 1A). Mark-release-recapture 4 (MRR 4) was a single linear release of males from a prototype mechanical device used in Crawford et al. [39], on the eastern side of the road, from north to south (Fig 1A).

## Trapping arrays and recaptures

The study site contained 83 Biogents Sentinel traps without lures (BGS; Biogents GmbH, Regensburg, Germany) set with the goal of distinguishing landscape characteristics that affected the movement of males through blocks and across movement barriers such as roads (Fig 1). To do this, one trap was placed close to each chosen dwelling, one in the backyard and, where possible, one near the forested area adjacent to the residential area. Additional traps were placed at dwellings across the road from the release sites to monitor movement across a known dispersal barrier. For MRR 1 and 2, traps were turned on after 24 hours to allow for mixing of marked males with the wild population. For MRRs 3–6, traps were turned on two hours post-release.

All traps were serviced daily throughout each MRR experiment until no marked males were present in the sample. Captured adult *Ae. aegypti* were stored at ~4˚C for transfer to the laboratory for identification, after which both males and females were processed for the presence of rhodamine B following the methods of Johnson et al. [31]. Females were considered to be inseminated by a released marked male if rhodamine B was observed in the bursa, spermathecae or both. Females were considered to have mated with a wild, unmarked male if sperm, visualised by DAPI staining, was present in the bursa, spermathecae or both in the absence of rhodamine B.

## Determination of biological parameters, statistical analysis and dispersal kernel framework

For all experiments the probability of daily survival (PDS) was estimated by regressing $\log_{10}$ $(x+1)$ the number of recaptured males against days since release, where the antilog$_{10}$ of the

regression slope was the PDS [40]. The ALE was calculated from the PDS as $1/-\log_e PDS$ [41]. The Lincoln Peterson Index with Chapman modifier was used to estimate population size [42]. Where $\hat{N}$ is the size of the population, $n_1$ is the number of marked animals released into the population, $n_2$ the total number of individuals captured (marked and unmarked), and $m_2$ the total number of marked individuals recaptured:

$$\hat{N} = \frac{(n_1 + 1)(n_2 + 1)}{(m_2 + 1)} - 1$$

To account for the assumption that marked insects become completely mixed within the local population, only males from the release block and from multi-point (MRR 3 & 5) or the linear releases were selected for analysis (MRR 4). To account for the low survival of released *Ae. aegypti* males, only marked males captured the days before the ALE (rounded to the nearest whole number) of each experiment were selected [42]. The remaining assumptions of the LPI [8] were met with reasonable certainty which include: 1) the mark should not affect insects, 2) sampling is random with respect to marked insects, 3) samples are measured at discrete time intervals in relation to total time, and 4) the population is not unduly influenced by immigration or emigration during the period of study.

Traditional methods of MDT were calculated using the methods of Lillie et al [29] and Morris et al [28] where annuli are drawn around the release point to estimate dispersal distances:

$$\mathrm{MDT} = \frac{\sum \textit{Estimated recaptures [ER]} \times \textit{median distance for each annulus (for all annuli)}}{\textit{Total number of ER}};$$

and then a correction factor (CF) is applied to accommodate unequal trapping densities

$$\mathrm{ER} = \frac{\textit{Number of observed recaptures in an annulus}}{\textit{CF} \div \textit{number of traps in the annulus}};$$

where:

$$\mathrm{CF} = \frac{\textit{Area of the annulus} \times \textit{the total number of traps}}{\textit{Total trapping area}}$$

Flight range (FR) of male movement was estimated from the linear regression of the cumulative ERs for each annulus (x axis) on the $\log_{10}$ (y axis) as the value of the y axis at 50% ($FR_{50}$) and 90% ($FR_{90}$), of the largest value of the x axis [13]. We introduce the concept of mean insemination distance (MID) by modifying the above methods of Lillie et al [29] and Morris et al [28] by estimating the mean distance over which rhodamine B inseminated females were captured during each experiment.

We then compared traditional MDT estimates with the spatially and temporally evolving isotropic kernel (STEIK) framework developed by Trewin et al. [19]. The STEIK framework uses an isotropic Gaussian diffusion model with kernels defined as a temporally-evolving probability density function (PDF) over two-dimensional space [19]. Here the probability of a mosquito being trapped per unit area is a function of the distance from the release location and the time since release [19]. Multi-point releases treat the trapping intensity at each site as a finite mixture of the dispersal kernels from each release point, so the unknown latent variable of release location for each trapped mosquito has been integrated out of the probability density function used for the likelihood equation. For multi-point releases we divided the total number of mosquitoes released evenly between release points. For STEIK estimates of 50% and 90% FR, quartiles of simulated kernel distributions were calculated from parameter estimates relevant to average lifetime and the standard deviation of the isotropic kernel for each experiment.

To facilitate the use of our STEIK framework by experimentalists, we have stored male recapture data and R code at https://github.com/dpagendam/MRRk [43] and at the CSIRO Software Collections under an Open Source Software Licence [44]. Raw mosquito and trap data are available at CSIRO Data Collections under a Creative Commons 4.0 Licence [45].

Mating competitiveness was estimated using the methods of Reisen et al. [46] where the mating competitiveness (C) was calculated from the number of unmarked males among all males (w); and the number of unmarked male matings among all determined matings (f) where:

$$C = \frac{w}{(1-w)} \times \frac{(1-f)}{f}$$

The variation and test for significance between experiments was via chi-squared test following Grover et al. [47]. To examine differences in the daily proportion of rhodamine B inseminated females with the total number of mated females between seasons, we used a mixed effects, logistic regression model with a binomial distribution and logit link function. Fixed effects included season and release type (multi-point vs point release) with a random effect of experiment number. The same model framework was used to examine differences in the total daily proportion of mated females (both rhodamine B and wild mated) between seasons and the daily proportion of wild-type mated females with total females captured. The Akaike information criterion (AIC) was used to selecte the most parsimonious model. Odds ratios (OR) were calculated for coefficients exhibiting significant differences in proportions. The R package 'glmmTMB' [48] was used for all mixed effects models and the packages 'DHARMa' [49] was used for model diagnostics and 'ggplot2' [50] for visualisations. To look for collinearity in predictors, correlations were examined using the R package 'corrplot' [51]. The Wall-Raff rank sum test of angular distance from the R package 'circular' was used to compare similarities between wind direction and male recapture angle [52]. To compare whether rhodamine B and wild males and wild female *Ae. aegypti* were more likely to be captured by BGS traps at certain locations (house, backyard or forest), we used contingency table analysis with odds ratios calculated via the R package 'epitools' [53]. All analyses were performed using R version 3.5.3 [54]. All landscape maps were digitized by outlining landscape features (houses, roads, blocks, river) in Google Earth [55] and modified in ArcGIS Desktop [56] and provided as a layer file [38]. Two-dimensional kernels were output as images from an R density function where the mean was equal to zero and the spread equal to the time dependent standard deviation of the kernel. Kernels were then overlaid on maps to scale in standard image editing software.

## Results

### Population statistics

Environmental conditions varied considerably both before and during each season. The most notable difference being total rainfall two weeks before each season, with combined totals of 65.8 mm and 762 mm falling before MRRs in season 1 and 2, respectively (S1 Table) [36]. Mean daily minimum and maximum temperatures during the study periods varied between 15.1˚C and 31.9˚C, and the mean relative humidity at 09:00 and 15:00 hours varied from 85.7% (SD ± 6.8) and 53.8% (SD ± 6.8), respectively (S1 Table). Wind direction was predominantly from the southeast (S1 Fig) and was significantly different to the daily mean angle of movement for male *Ae. aegypti* from single point releases from season 1 ($\chi^2 = 9.05$, $P < 0.005$) and season 2 ($\chi^2 = 12.06$, $P < 0.001$) (S2 Fig). Total rainfall for the two weeks before and during releases varied from 0 mm in November to 344 mm in February (S1 Table) [36].

A total of 313 (4.1%) marked male *Ae. aegypti* were recaptured from a total of 7,713 released into the South Innisfail landscape. Recapture rates for rhodamine B marked *Ae. aegypti* males

**Table 1. Mating and recapture results from individual rhodamine B marked *Aedes aegypti* experiments in South Innisfail, Australia.** Wild females were examined for rhodamine B insemination by fluorescent microscopy.

| MRR | Trapping Period (Days) | Rho B Males Released | Trapped *Aedes aegypti* individuals | | | | | | | Mating Competitiveness |
| --- | --- | --- | --- | --- | --- | --- | --- | --- | --- | --- |
| | | | Rho B Marked Males (%) | Wild Males | Total Males (% Rho B) | Rho B Mated Females (%) | Females Mated by Wild Males (%) | Unmated Females | Total Females (% Mated) | |
| 1 | 8 | 1228 | 17 (1.4) | 72 | 89 (19.1) | 22 (18.9) | 61 (52.6) | 33 | 116 (71.6) | 1.53 |
| 2 | 9 | 1485 | 11 (0.7) | 51 | 62 (17.7) | 28 (17.4) | 82 (50.9) | 51 | 161 (68.3) | 1.58 |
| 3 | 7 | 1240 | 12 (1.0) | 28 | 40 (30.0) | 9 (12.2) | 30 (40.5) | 35 | 74 (52.7) | 0.7 |
| 4 | 7 | 1250 | 42 (3.4) | 130 | 172 (24.4) | 13 (4.4) | 127 (43.2) | 154 | 294 (47.6) | 0.32 |
| 5 | 7 | 1250 | 130 (10.4) | 57 | 187 (69.5) | 13 (5.4) | 123 (51.5) | 103 | 239 (56.9) | 0.05 |
| 6 | 6 | 1250 | 102 (8.1) | 42 | 144 (70.8) | 12 (10.6) | 57 (50.4) | 44 | 113 (61.1) | 0.09 |

varied during individual experiments (0.7% to 10.4% recaptured), with the mean number of recaptures generally increasing on day two post release (mean = 24.5, SD ± 22.0), before decreasing (Table 1, Fig 2). Recapture success was highly variable between the dry and wet seasons, averaging 1.0% (SE ± 0.2) and 7.3% (SE ± 4.3) recaptured, respectively (Table 1). Maximum time to recapture (the period between release and last date of a marked individual captured) varied from three to seven days across all experiments (Table 2). There were significantly more wild-type male and female *Ae. aegypti* caught daily in season 2 than in season 1 (Fig 3, $F_{(2,41)}$ = 18.01, P = <0.001).

Estimates of survival and ALE after release varied considerably for each experiment (Table 2). When combining data across all MRR experiments, we estimated the ALE as 1.69 days (PDS = 0.55). The maximum ALE observed in an individual experiment was 4.9 days (PDS = 0.82) during MRR 3, and a minimum of 0.47 days (PDS = 0.12) during MRR1

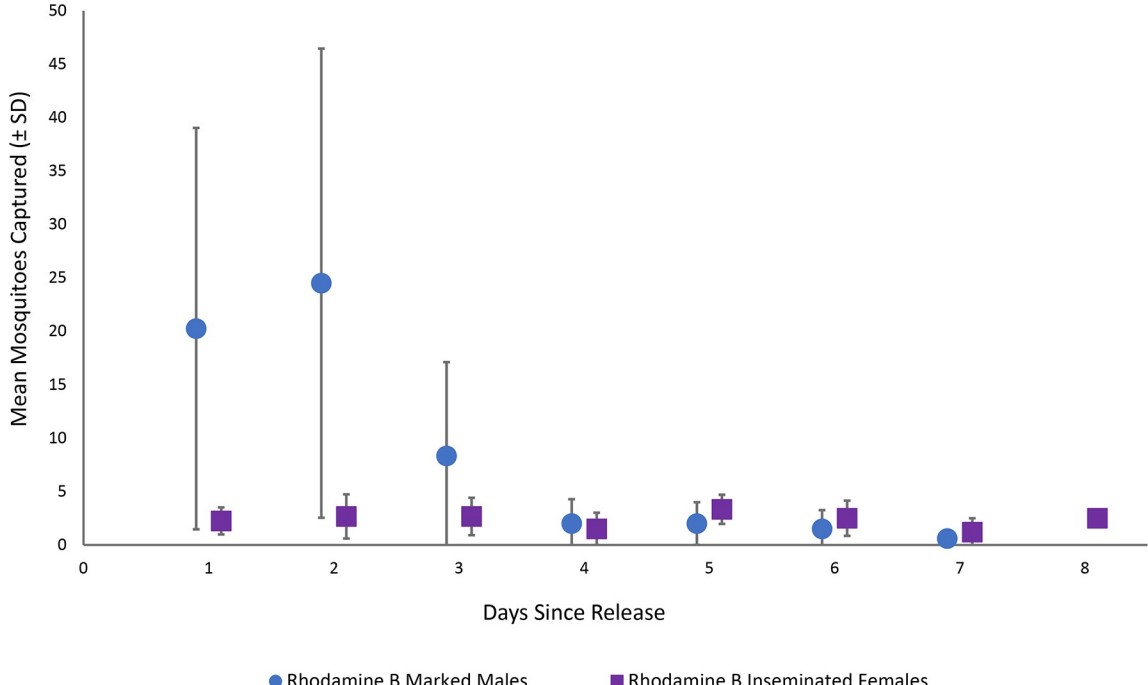

**Fig 2. Daily mean recapture rate (±SD) of rhodamine B marked male (blue circles) and inseminated female *Aedes aegypti* (purple squares) in South Innisfail, Australia.** Data aggregated across all mark-release-recapture experiments.

**Table 2. Probability of daily survival, average life expectancy and population estimates from rhodamine B marked *Aedes aegypti* during six mark-release-recapture experiments in South Innisfail, Australia.** Wild male population sizes were estimated via the Lincoln Peterson Index [8] and probability of daily survival and average life expectancy via the methods of Gillies [40] and Niebylski and Craig Jr [41].

| MRR | Period | Release Type | Maximum Time to Recapture (Days) | Probability of Daily Survival | Average Life Expectancy (Days) | Male Population Estimate* | Lower Estimate | Upper Estimate |
|-----|--------|--------------|-------------------------------|------------------------------|-------------------------------|--------------------------|----------------|----------------|
| 1 | Dry Season | Single Point | 3 | 0.12 | 0.47 | na | na | na |
| 2 | Dry Season | Single Point | 6 | 0.73 | 3.23 | na | na | na |
| 3 | Dry Season | Multi-Point | 6 | 0.82 | 4.94 | 3,100 | 2,021 | 5,667 |
| 4 | Wet Season | Linear | 7 | 0.81 | 4.86 | 4,107 | 2,790 | 6,551 |
| 5 | Wet Season | Multi—Point | 7 | 0.56 | 1.73 | 1,418 | 1,335 | 1,582 |
| 6 | Wet Season | Single (Night) | 6 | 0.54 | 1.64 | na | na | na |

* Male population was estimated using males collected in traps within the release block, during multi-point and linear release types. To account for low survival, this estimate only took into consideration those wild-type and rhodamine B marked males collected within the period up to the ALE (rounded to the nearest whole number).

(Table 2). Estimated wild male population sizes ranged from 1,418 (Table 2; 95% CI = 1,335–1,582) to 4,107 (95% CI = 2,790–6,551), which would represent 53–152 males per premises within the release block (Table 2).

## Mating interactions

The daily proportion of rhodamine B inseminated females tended to be low but consistent across all experiments (Figs 2, 3 and 4). Likewise, the proportion of all inseminated females (wild and rhodamine B) remained relatively constant across experiments, while total rhodamine B inseminations tended to vary relative to season and total females captured. Between 25–52% of females captured per experiment were uninseminated (Figs 3 and 4). There was no significant difference in mating competitiveness of marked and wild-type males ($\chi^2$ = 2.04, df = 5, P > 0.05, Table 1). The mixed effects logistic regression model revealed the daily proportion of females inseminated by rhodamine B marked males was significantly higher in season 1 than season 2 (Z = -2.81, df = 37, $P$ < 0.005) and in single point than multi-point releases (Fig 5; Z = -2.39, df = 37, P = 0.017), respectively. This equated to a ~54% and ~48% decrease in the daily odds of a female being inseminated by rhodamine B marked males during season 2 (Table 3; OR = 0.46, 95% CI Low = 0.27, High = 0.79) and during linear releases (OR = 0.52, 95% CI Low = 0.30, High = 0.89), respectively. However, when the same statistical model was used to examine the proportion of daily total mated females to total female mosquitoes captured, there was a significantly higher proportion during single point releases (Z = -2.876, df = 38, $P$ < 0.004) but not season (Z = -1.20, df = 38, $P$ = 0.23). Furthermore, there was no significant relationship between the proportion of wild type male *Ae. aegypti* inseminations to total females captured between seasons (Z = -0.39, df = 38, $P$ = 0.93).

## Movement estimates

Males moved rapidly through the environment and were observed up to 422m from the single point of release. Movement was highest during the night release experiment for all metrics calculated (MRR 6; Table 4). The multi-point isotropic Gaussian kernel framework provided

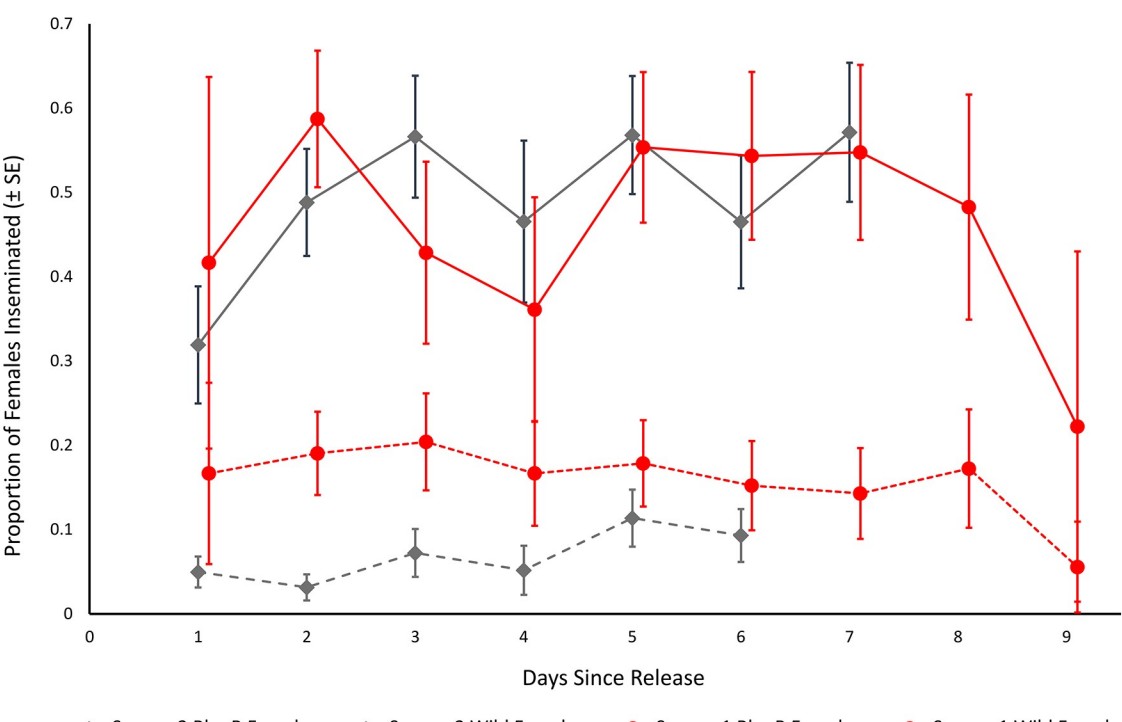

**Fig 3. Mean daily proportion (SE) of captured *Aedes aegypti* females inseminated by wild or rhodamine B marked males.** Differences between time of year are indicated by early summer (season 1, red circles) and late summer (season 2, grey diamonds).

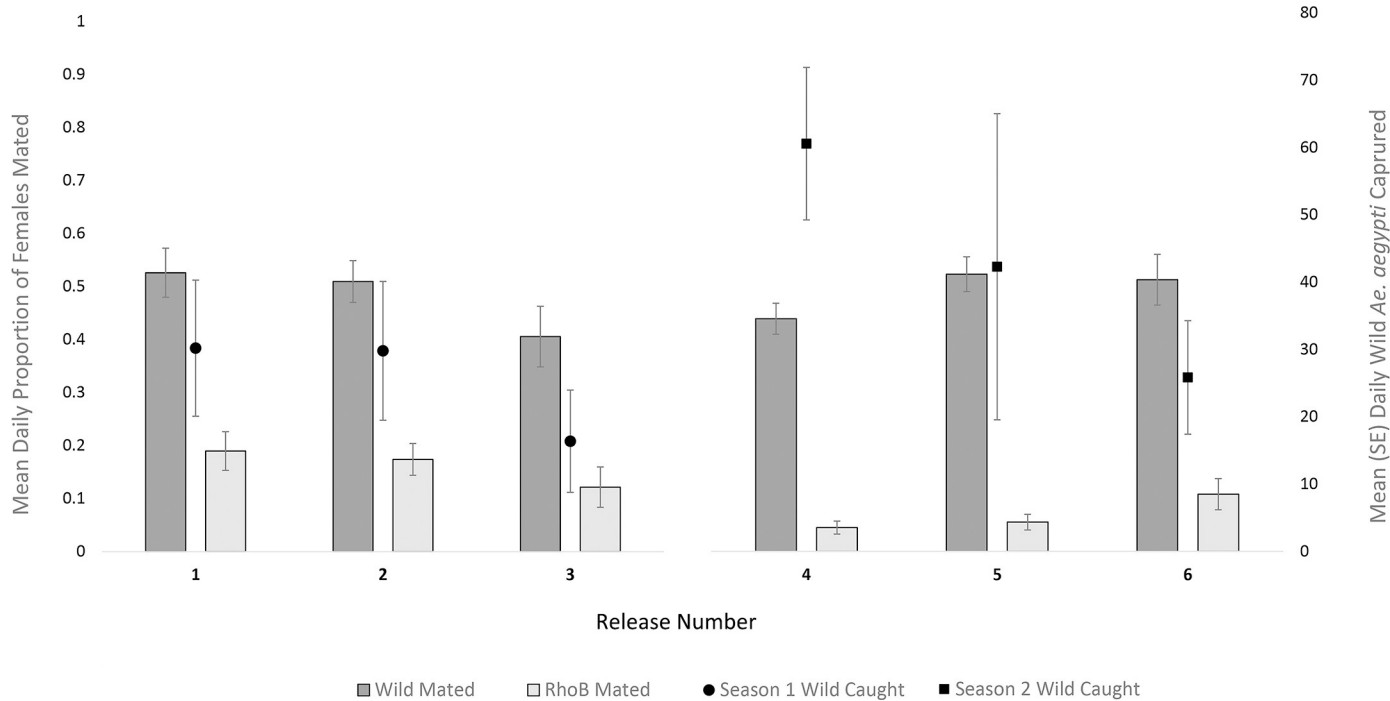

**Fig 4. A comparison of mating rate with wild *Ae. aegypti* captured during six mark-release-recapture experiments in Innisfail, Australia.** Primary axis indicates the mean daily proportion (± SE) of wild and rhodamine B inseminated female *Ae. aegypti*. Second axis indicates the mean daily total (male and female) number (± SD) of wild male and female *Ae. aegypti* captured during each experiment.

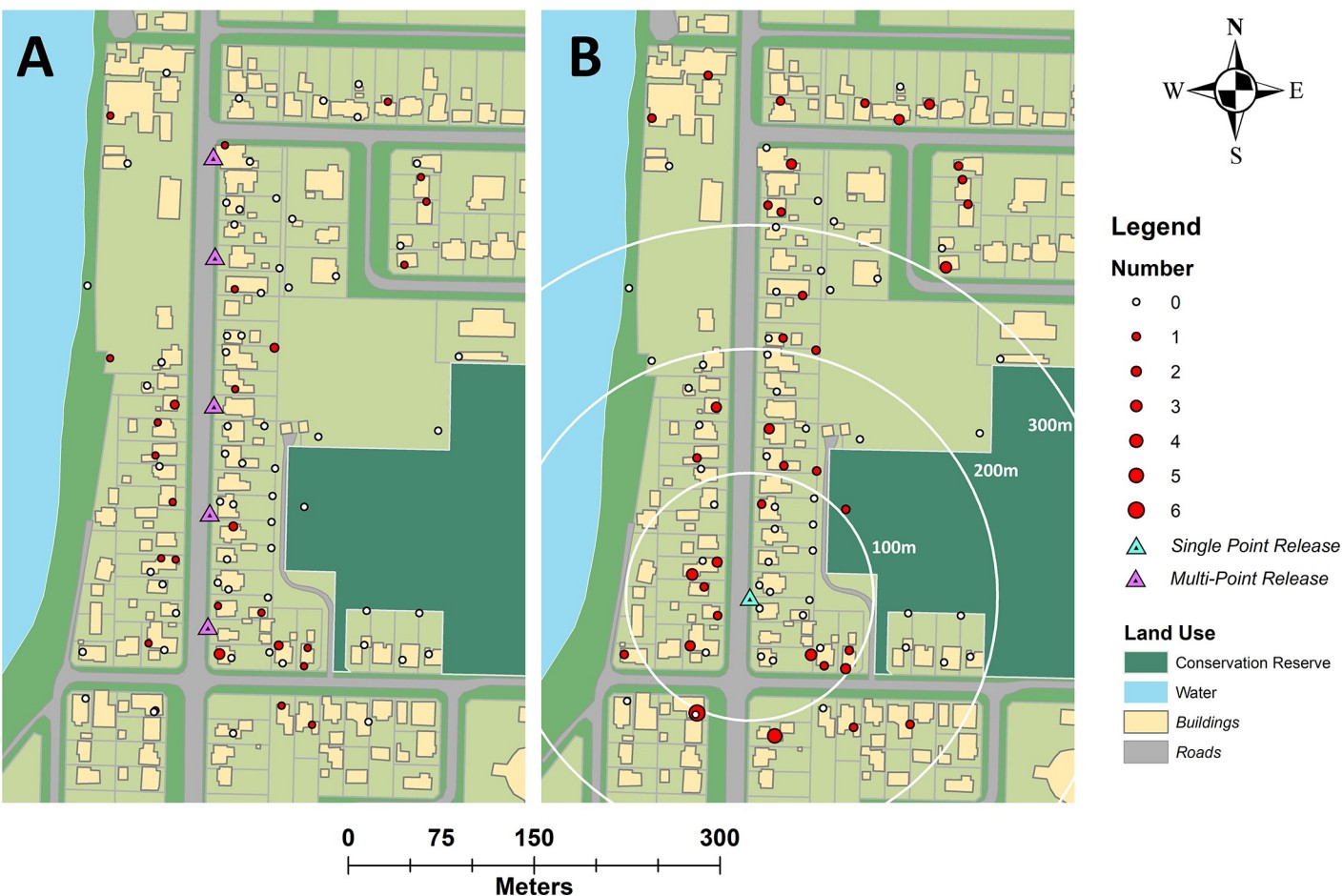

**Fig 5. Total rhodamine B inseminated *Aedes aegypti* females captured during multi-point (5A) and single point releases (5B).** Size and colour of circles indicates the total females caught in an individual trap (see key). Release points for each release type are indicated by blue triangle (single point) and purple triangles (multi-point). Landuse basemap digitized manually [38].

MDT estimates comparable to those of Lillie et al. [29]. When MDT is estimated over the entire period of each experiment, the STEIK generally estimated greater movement distances than the traditional method. For example, males were estimated to travel a mean of 295.2 m ($FR_{50/90}$ 248/480 m) compared with 451 m ($FR_{50/90}$ 335/873 m) by traditional and STEIK

**Table 3. Results of the mixed effects logistic regression model on daily proportion of females mated.** Results show the effect of release type (single or multi-point) and season (early and late summer) on the daily proportion of rhodamine B inseminated females captured.

| Fixed Effects | | β | SE β | z value | P | odds ratio |
|---|---|---|---|---|---|---|
| Constant | | -1.49 | 0.15 | -9.82 | <0.001 | 0.23 |
| Release Type | | -0.66 | 0.28 | -2.39 | 0.017 | 0.52 |
| Season | | -0.77 | 0.28 | -2.81 | 0.005 | 0.46 |
| Random Effect | | $\sigma^2$ | SE β | | | |
| Experiment | | <0.001 | <0.001 | | | |
| Overall Model Evaluation | Test | | | | | |
| | Kolmogorov -Smirov | | | D = 0.149 | P = 0.291 | |
| | Overdispersion | | | Ratio = 0.878 | P = 0.332 | |

**Table 4. Movement estimates male *Aedes aegypti* from six mark-release-recapture experiments in South Innisfail, Australia.** Spatially and Temporally Evolving Isotropic Kernel (STEIK) framework compared with the traditional annulus-based method. The STEIK framework allows for estimates of movement based on average life expectancy and from multiple release points (MRR 3 & 5).

| MRR | Maximum Distance Travelled (m) | Max Time (d) | Traditional MDT (m)[β] | Flight Range 50/90% (m) | STEIK MDT (m)[*] | Flight Range 50/90% (m) | Average Life Expectancy (d) | STEIK ALE-MDT (m)[¥] | Flight Range 50/90% (m) |
|---|---|---|---|---|---|---|---|---|---|
| **1** | 187 | 3 | 126.7 | 118/261 | 95.4 | 68/172 | 0.5 | 51.8 | 43/109 |
| **2** | 282 | 7 | 233.6 | 202/342 | 278.3 | 195/499 | 3.2 | 179.6 | 146/367 |
| **3** | na | 6 | na | na | 110.4 | 76/205 | 4.9 | 95.0 | 75/195 |
| **4** | na | 7 | na | na | na | na | 4.9 | na | na |
| **5** | na | 7 | na | na | 184.8 | 132/333 | 1.7 | 93.5 | 76/187 |
| **6** | 422 | 6 | 310.5 | 261/508 | 462.8 | 326/837 | 1.6 | 246.7 | 199/501 |
| **Mean** | 297 | 6 | 295.2 | 248/480 | 451.7 | 335/873 | 1.6 | 240.8 | 195/497 |

[β] Calculated using the traditional methods of Morris, Larson and Lounibos [28] and Lillie, Marquardt, and Jones [29] over the period of study.

[*] Method for MDT uses a random lifetime generated from the maximum time to recapture.

[¥] Method samples from one million lifetimes with a distribution equivalent to the estimated daily survival rate from the field.

methods, respectively (Table 4). The highest STEIK MDT estimate was 462.8 m (Fig 6; $FR_{50/90}$ 326/837 m) compared to a traditional estimate of 310.5 m during MRR6 (Table 4; $FR_{50/90}$ 261/508 m). Male MDT during the multi-point releases in MRR3 and MRR5 were unable to be estimated using the traditional method of Lillie et al. [29], as it relies upon a discrete point of release. However, the multi-point, STEIK framework estimated an MDT of 110.4 m ($FR_{50/90}$ 76/205 m) and 184.8 m ($FR_{50/90}$ 132/333 m) for MRR3 and 5, respectively (Fig 7 and Table 4). The MDT for MRR4 was unable to be calculated by either traditional or multi-point isotropic

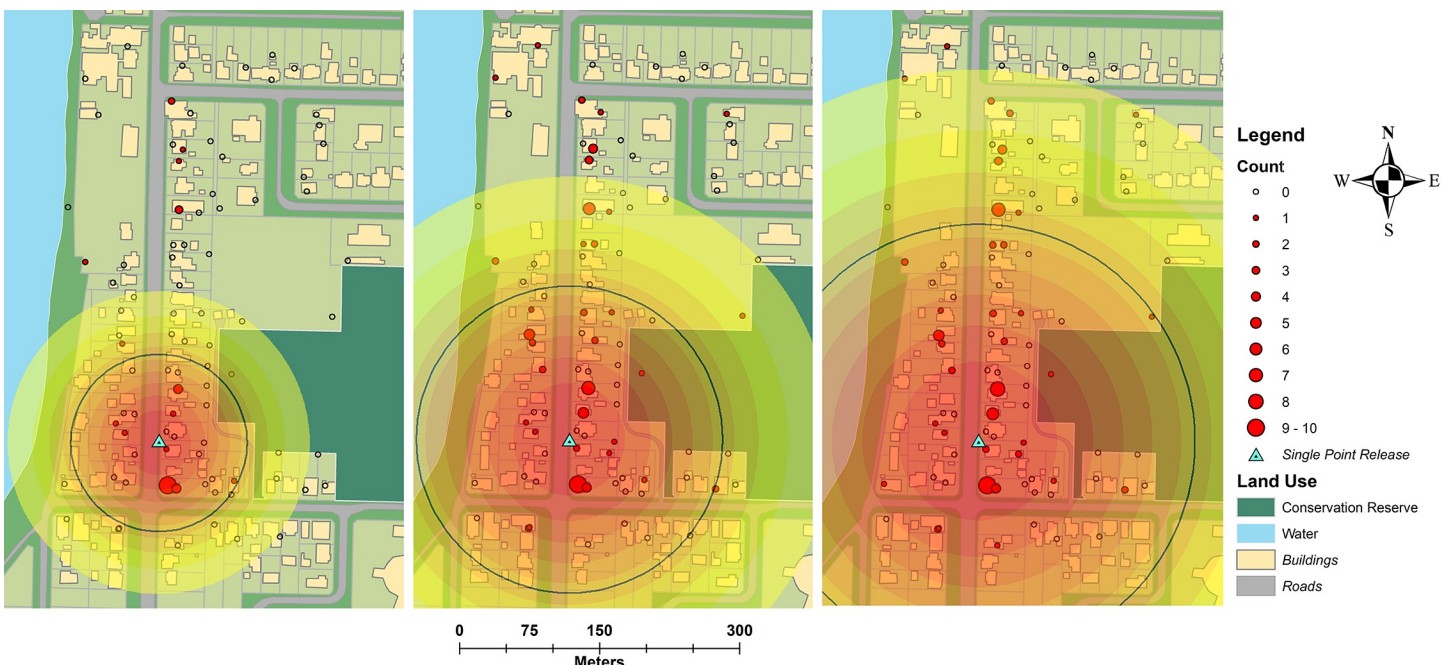

**Fig 6. Single point estimates of *Aedes aegypti* movement during experiment six which apply the STEIK framework.** Concentric circles from release point are density estimates of marked adult male *Aedes aegypti* one (A), three (B) and six (C) days post release. Black lines represent mean distance travelled over the time period. Landuse basemap digitized manually [38].

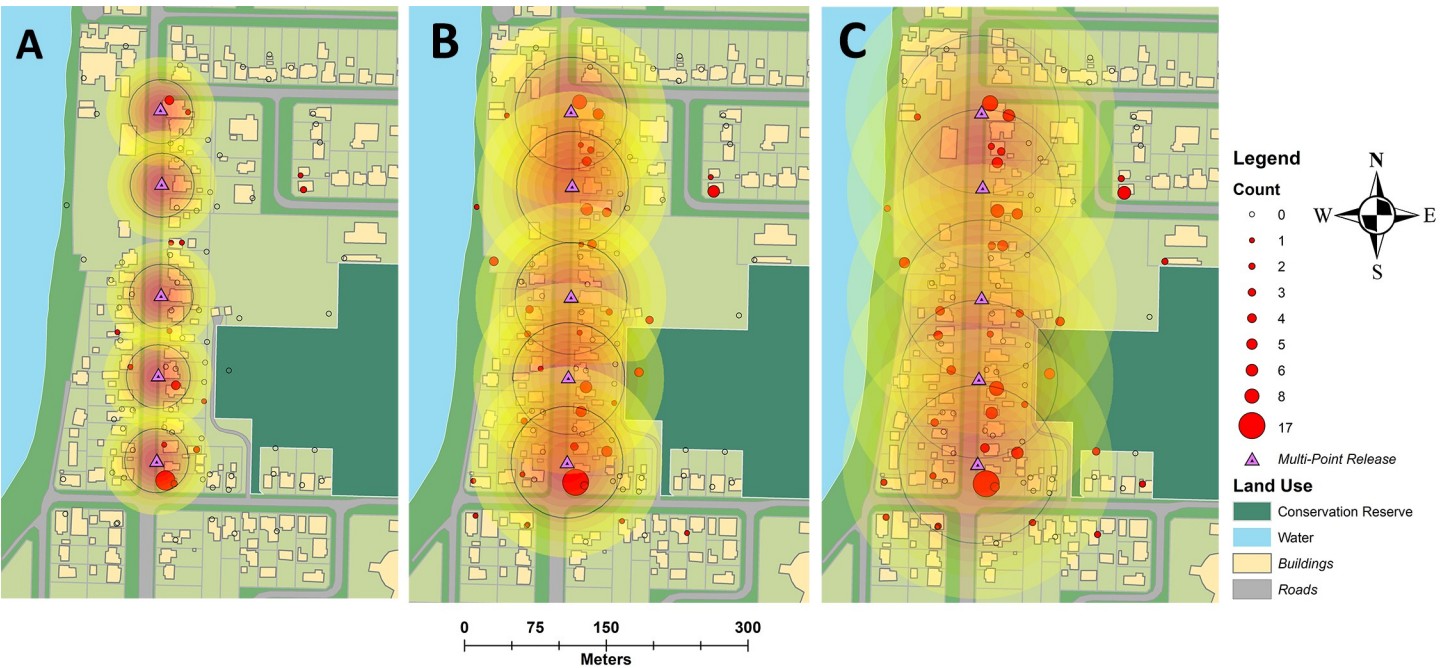

**Fig 7. Multi-point estimates of *Aedes aegypti* movement during experiment four which apply the STEIK framework.** Concentric circles from release point are density estimates of marked adult male *Aedes aegypti* one (A), three (B) and six (C) days post release. Black lines represent mean distance travelled over the time period. Landuse basemap digitized manually [38].

kernel methods as this release did not use discrete points as male releases were by a moving vehicle and regarded as linear. Using average life expectancy to estimate a distribution of survival, estimates of MDT were generally lower or equal to those estimated by the isotropic kernel or traditional methods that incorporate the maximum time to recapture (Table 4). The maximum distance over which rhodamine B inseminated females were captured was greater than marked males in MRRs 1 & 2, with marked females caught at the maximum distance of our trapping network. Total and daily female MID were considerably higher than male movement calculations (Table 5).

## The role of urban landscape features

Contingency table analysis revealed Rhodamine B marked male *Ae. aegypti* were two times more likely to be captured in BGS traps situated at houses than in backyards or forests ($X^2$ = 10.06, df = 2, N = 564, Backyard OR = 0.47, 95% CI = 0.28–0.77, Forest OR = 0.46, 95% CI = 0.18–1.19, $P < 0.007$). Likewise, wild-type female *Ae. aegypti* had twice the likelihood of

**Table 5. Mean Insemination Distances (MID) estimates adapted from traditional annulus-based methods.** Estimates give an estimation of the distance at which inseminated females are then captured over time.

| MRR | Maximum Insemination Distance (m) | Maximum Time to Recapture (days) | Total MID [β] (m) | Daily MID [β](m) | | | | | | |
|---|---|---|---|---|---|---|---|---|---|---|
| | | | | Day 1 | Day 2 | Day 3 | Day 4 | Day 5 | Day 6 | Day 7 |
| **1** | 425 | 7 | 387.9 | na | 283.8 | 303.6 | 355.6 | 298.0 | 89.3 | 300.7 |
| **2** | 411 | 6 | 337.6 | na | 215.8 | 264.4 | 96.9 | 285.3 | 227.3 | na |
| **6** | 425 | 6 | 374.7 | 125.0 | 342.7 | 303.6 | na | 156.9 | 316.8 | na |

[β] Mean Insemination Distance (MID) calculated using the traditional methods of Morris, Larson [28] and Lillie, Marquardt, and Jones [29].

being captured in traps around houses than in backyards or forests ($X^2$ = 9.81, df = 2, N = 564, Backyard OR = 0.51, 95% CI = 0.33–0.80, Forest OR = 0.57, 95% CI = 0.24–1.37, $P < 0.007$). Wild-type males had a similar likelihood of being captured in traps around houses and forests ($X^2$ = 19.00, df = 2, N = 564, Forest OR = 1.09, 95% CI = 0.50–2.49) but half the likelihood of being captured in backyard traps (OR = 0.47, 95% CI = 0.32–0.69).

## Discussion

*Aedes* control is entering a new era of area-wide, rear-and-release control strategies, where accurate measurements of population parameters will be essential for operational success [39,57]. Incompatible insect and sterile insect field interventions rely on several important biological parameters that determine the effectiveness of mass-reared male mosquitoes. Sterile insect interventions are reliant on the ability of mass-reared males to survive, seek out and successfully mate with available wild females. As female *Ae. aegypti* are typically thought to mate once during their lifetime, sterilization is reliant upon releasing enough mass-reared males (over-flooding the population) to outcompete wild males. Here we performed the first large-scale field releases of rhodamine B marked male *Ae. aegypti* into a wild population of mosquitoes. Our goal was to measure the basic biological parameters that govern movement, survival and mating interactions of mass-reared males within this population.

Competition for mates between released and wild type male mosquitoes is difficult to discern within a heterogeneous population. Here, rhodamine B was used to mark the body and seminal fluid of male *Ae. aegypti*, enabling us to measure mating interactions with the wild female population and compare mating competitiveness between released and wild mates. Marked males showed no significant differences in mating competitiveness across experiments. This made no difference to the proportion of all inseminated females throughout each experiment (both rhodamine B and wild mated), and this effect was consistent across seasons. As release numbers were relatively similar across all experiments, results suggest that the proportion of total mated females remains relatively static regardless of the number of additional males added to a population and sterile to wild-type mating ratios will be proportional to the number of each in the population [58]. Wild females are likely available for mating immediately upon release of marked males, with ~40% of females unmated by marked or wild-type males at any given time. This suggests that females emerged and became available for mating at a relatively constant rate throughout the study. As *Ae. aegypti* males are limited by the rate at which they can mate during their lifetime [59], ensuring a constant supply of males into the landscape should be the priority of sterile male programs. Multiple releases of males will be required each week to ensure a constant and efficient overflooding ratio, a number that is highly reliant on the fitness of released males. Doing so will ensure a saturated landscape where incompatible males sterilize females as soon as they emerge while also minimising the effects of immigration. If sterile males are not constantly present within the landscape, then programs will tend to be inefficient with higher costs and lower suppression than expected over time, similar to normal larviciding programs.

Our results support recent studies which show male *Ae. aegypti* are capable of moving further through an urban landscape than historical observations suggest. Variations in MDT are likely due to the chosen mark, collection method and landscape configuration [12]. These results have implications for how, where and when mosquito programs release males into local landscapes during SIT/IIT interventions. Here, we show that male *Ae. aegypti* have the potential for long distance movement despite a short lifespan, supporting recent dispersal studies using stable isotopes [17–20]. During single point releases marked males were recaptured at distances greater than 350 m and 400 m on day one and two after release, respectively,

suggesting rapid movement when compared to females of the species. Furthermore, rhodamine B inseminated females were captured at the extent of the trapping array, and we observed a MID greater than the male MDT. This suggests marked males had an effect on the female population greater than the distance over which they were captured, likely due to female movement. Both isotropic and traditional MDT measurements of male movement (between 95 m and 462 m and across movement barriers) support the general observation that the rate of male movement was greater than those traditionally recorded for females of the species in Australian landscapes. This could suggest that gene flow within an isolated population may be related to male spread as female movement is typically observed at less than 100 m over a lifetime [10,13]. This difference may be due to the different biological requirements of each sex, with females requiring a blood source, a resting place and oviposition site that may reduce dispersal, whereas the short life expectancy of males may have them constantly searching for virgin females. A greater understanding of dispersal patterns in local environments will result in maximum sterile male coverage across a landscape, and lead to greater mating success, the primary goal of area-wide release programs.

Although single point MRRs are ideal for measuring distance travelled over time, the utilization of measurements from multi-point releases are essential to optimizing area-wide releases. The STEIK estimates of movement used in this study [19,43] benefited from the addition of temporal data and tended to produce larger estimates than those calculated through traditional methods. Interestingly, our kernel estimates of male MDT were similar to Marcantonio et al [18] over the period of one week. However, static kernels such as this and Winskill et al [23] are based on seed dispersal where individual movement is not estimated once collected in traps [19]. The STEIK method has the additional benefit of estimating movement throughout a life-time as a temporally evolving kernel [19]. While greater male MDT was observed in single point rather than during multi-point releases, this is likely a function of both the release strategy (males being released across many traps) and a lack of clarity from which release sites captured mosquitoes had originated from (as a result of the single mark method employed). The latter point is important since our model considers that it was less likely that a mosquito caught close to one release point had travelled there from a distant release point, even though this could be the true movement pattern. During multi-point releases with a single marking type, the cumulative probability would therefore tend to be conservative when estimating MDT. As such, single point and multi-point releases cannot be easily compared when only a single marking type is used, however, this could be improved by incorporating different marking types.

It is widely known that sterilization through transgenic and radiation approaches impose a fitness burden on released males [60,61]. Because of this, modern SIT programs have regularly failed due to lack of knowledge on sterile male performance post-release [57,62,63]. Although we observed considerable differences in ALE between experiments, our results confirm the relatively short lifespan of male *Ae. aegypti* compared to that of females. It is estimated that female *Ae. aegypti* adults live on average for between five and nine days in the field, with survival curves showing enough females survive the extrinsic incubation period to transmit pathogens [13,64–66]. The differences between male and female lifespan has implications for scheduling the production and release of sterile male mosquitoes to achieve adequate overflooding of a wild population. If males used in a *Wolbachia* based IIT approach only survive for three to five days post-release, then releases will need to occur over regular intervals to ensure mass-reared males outcompete wild males and achieve the desired level of suppression.

The LPI is a closed population method used historically for estimating population sizes during mark-release-recapture experiments. Ratios employed by this index compare a simplistic relationship between the total number of individuals captured, with the numbers released and

then recaptured [8]. Our results confirm the difficulty of using this index to accurately estimate mosquito population sizes, particularly due to a lack of biological information during releases. For instance, it is enforce the strict almost certain that the assumption of a closed mosquito population closed and that the probability of recapture remains constant over space and time are incorrect. This is because new individuals are constantly being recruited into a population during favourable periods but are also dying constantly due to short lifespans, in heterogeneous spatial distributions across a landscape. When we observe proxies for these rates, such as estimates of average lifespan and mating rates, our faith in the estimates obtained from the LPI may diminish, depending upon to what extent the above assumptions may be violated.

Inaccuracies in LPI estimates are most evident when we compare the populations in our study (Table 2) with the total male and female captures between seasons (Fig 4). The larger mean daily trapping rate during season 2 would suggest a population 1.5–2 times larger than during season 1. However, the LPI estimate does not reflect this trend, with results suggesting similar sized populations between MRR 3 and 4, but lower in MRR 5. Although inaccuracies in our population estimates are reflected by large confidence intervals, it is likely that even the lowest confidence intervals are unreflective of the male *Ae. aegypti* population in the field, which when divided by the number of premises within the release block suggest 49–75 males per house. Extensive *Ae. aegypti* research in north Queensland suggests this population estimate is much higher than would be expected in the region, given the number of adults typically observed per premises during the wet season is ca. 10 [67,68]. Furthermore, *Ae. aegypti* is an urban container inhabiting mosquito and populations are correlated with precipitation and temperature [67,69]. The lack of rainfall during the two weeks leading up to MRR 3, (0.8 mm) would suggest a considerably lower population when compared to MRR4 (242 mm). However, this was not the case when populations were estimated by the LPI and could be the result of low re-captures. Future experiments that combine both the LPI and the insemination rate (or competitiveness) could overcome inaccuracies associated with low recapture rates and may lead to improved estimates of mosquito populations in the field.

Until recently, a lack of male *Ae. aegypti* movement studies resulted in a deficiency of knowledge on individual behaviour and their ability to navigate through landscapes. This study placed traps in three major microhabitats, house, backyard and forest, to observe how urban landscapes in north Queensland impact the movement of male *Ae. aegypti*. During both single and multi-point releases males tended to be recaptured within the same block they were released, and wind had no influence on the direction of movement. These findings support previous studies that suggest physical barriers influence movement between residential blocks [10,19]. However, roads or wind direction did not totally prevent male movement or inseminated females moving into surrounding blocks, which suggests the open areas of Innisfail (such as roads or open grassy areas) provided minimal resistance. Interestingly, both marked and unmarked males and females were captured in traps lining forest lines, with the majority of marked males in MRR 1 caught in a forest trap directly behind a house at the single point release site. Forest traps also had the same likelihood of trapping wild males and females as traps situated in backyards. This observation supports previous studies that suggest that both male and female *Ae. aegypti* may instinctively move towards dark harbourage areas to seek shelter [70].

It is generally assumed that *Ae. aegypti* is most active during the daytime, as females show increased biting activity during the early morning and late afternoon [71]. However, when males were released at night, we observed higher recaptures with a greater total MDT and a rapid movement within the landscape when compared with daytime releases. It is possible that *Ae. aegypti* males move further through the night with higher humidity and lower predation playing less of a role than during daylight. Alternatively, males may have had orientational

problems finding resting sites, thus increasing movement across the landscape. Although we observed higher overall male movement during the night release, the lowest proportion of mated females during single-point releases was observed during this experiment. As the night release was not replicated caution should be assumed when interpreting these results and as such, additional studies are needed before firm conclusions can be drawn.

## Conclusion

The key to the next era of rear and release vector control will lie in the capacity of authorities to release competitive males that disperse widely, survive for greater periods and interact effectively with wild females. While the scientific literature contains extensive detail on female movement, there is relatively little quantitative information detailing the behaviour of wild-type male *Ae. aegypti*, and no studies exploring insemination rates in a field setting. Not only does rhodamine B provide new insights into male *Ae. aegypti* movement characteristics in urban landscapes, but additional information on how efficacious mass released male mosquitoes are at searching for and inseminating females in a wild population. The unique insights provided by our study into male *Ae. aegypti* biology will lay a foundation for designing and optimizing robust and effective male release strategies in the future and lead to a greater understanding of mating interactions in the wild.

## Supporting information

**S1 Table. Rainfall, mean daily minimum and maximum temperatures and mean relative humidity through the experimental periods.**
(DOCX)

**S1 Fig. Wind direction during experimental periods.**
(PDF)

**S2 Fig. Recaptured mosquitoes relative to wind direction during experimental periods.**
(PDF)

## Acknowledgments

We would like to thank Matt Bradford, Caleb Anning and Ben Purcell for assistance in trap placement, public engagement and mosquito collections. Also, to Helen Cook for her contribution and fearless work ethic in ensuring the community was successfully engaged and supportive during the extensive study period. We would like to acknowledge Kamran Najeebullah at CSIRO for wind direction plots relative to mosquito recaptures.

## Author Contributions

**Conceptualization:** Brendan J. Trewin, Daniel E. Pagendam, Brian J. Johnson, Nigel Snoad, Scott A. Ritchie, Bradley J. White, Sara Mitchell, Nigel W. Beebe.

**Data curation:** Brendan J. Trewin, Daniel E. Pagendam, Brian J. Johnson, Chris Paton, Nigel W. Beebe.

**Formal analysis:** Brendan J. Trewin, Daniel E. Pagendam.

**Funding acquisition:** Nigel W. Beebe.

**Investigation:** Brendan J. Trewin, Brian J. Johnson, Chris Paton, Nigel Snoad, Scott A. Ritchie, Kyran M. Staunton, Nigel W. Beebe.

**Methodology:** Brendan J. Trewin, Daniel E. Pagendam, Brian J. Johnson, Chris Paton, Scott A. Ritchie, Kyran M. Staunton, Bradley J. White, Sara Mitchell, Nigel W. Beebe.

**Project administration:** Nigel Snoad, Scott A. Ritchie, Nigel W. Beebe.

**Resources:** Nigel Snoad, Scott A. Ritchie, Nigel W. Beebe.

**Software:** Brendan J. Trewin, Daniel E. Pagendam.

**Supervision:** Nigel W. Beebe.

**Validation:** Brendan J. Trewin, Daniel E. Pagendam, Brian J. Johnson, Nigel W. Beebe.

**Visualization:** Brendan J. Trewin, Nigel W. Beebe.

**Writing – original draft:** Brendan J. Trewin, Daniel E. Pagendam, Nigel W. Beebe.

**Writing – review & editing:** Brendan J. Trewin, Daniel E. Pagendam, Brian J. Johnson, Chris Paton, Scott A. Ritchie, Kyran M. Staunton, Bradley J. White, Sara Mitchell, Nigel W. Beebe.

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
