## [Decision Letter · Decision Letter 0]

28 Dec 2020

Dear Dr Trewin,

Thank you very much for submitting your manuscript "Mark-release-recapture of male Aedes aegypti (Diptera: Culicidae): use of rhodamine B to estimate movement, mating and population parameters in preparation for an incompatible male program." for consideration at PLOS Neglected Tropical Diseases. As with all papers reviewed by the journal, your manuscript was reviewed by members of the editorial board and by several independent reviewers. In light of the reviews (below this email), we would like to invite the resubmission of a significantly-revised version that takes into account the reviewers' comments. 

We cannot make any decision about publication until we have seen the revised manuscript and your response to the reviewers' comments. Your revised manuscript is also likely to be sent to reviewers for further evaluation.

Sincerely,

Roberto Barrera, Ph.D.

Associate Editor

Eric Dumonteil

Deputy Editor

Reviewer's Responses to Questions

**Key Review Criteria Required for Acceptance?**

**Methods**

-Are the objectives of the study clearly articulated with a clear testable hypothesis stated?

-Is the study design appropriate to address the stated objectives?

-Is the population clearly described and appropriate for the hypothesis being tested?

-Is the sample size sufficient to ensure adequate power to address the hypothesis being tested?

-Were correct statistical analysis used to support conclusions?

-Are there concerns about ethical or regulatory requirements being met?

Reviewer #1: The MRR methods are well done and clearly presented. Several aspects should be considered:

1. LPI: Because male survival was so low, n1 should be adjusted to account for daily losses due to emigration and death. n2 is presumed constant with losses = additions.

2. Mating competitiveness. There are published methods to estimate male mating competitiveness [see Grover et al. 1976 Bull Ent Res 66: 469-480 and for aegypti Ent Exp. Appl. 20: 8-18; an example of their use is attached]. The authors have all the necessary data to estimate the performance of the released males and should provide these metrics as they will indicate the degree of 'over-flooding' necessary.

3. Dispersal. Comparing single vs transect releases was difficult because the authors could not tell from where males were released when recaptured during the transect release. This was discussed, but could be made more clear. 

4. Female age structure. It was not clear if the BGS traps collect mostly unfed/newly emerged females resulting in the very high proportion of uninseminated females reported. Perhaps the authors could briefly indicate the proportions of unfed, blood fed and gravid females collected in the Results section as this would affect the insemination rate? Was there any adult mosquito control that would eliminate the older portion of the wild population?

Reviewer #2: (No Response)

**Results**

-Does the analysis presented match the analysis plan?

-Are the results clearly and completely presented?

-Are the figures (Tables, Images) of sufficient quality for clarity?

Reviewer #1: The analyses and results were clearly presented.

Reviewer #2: (No Response)

**Conclusions**

-Are the conclusions supported by the data presented?

-Are the limitations of analysis clearly described?

-Do the authors discuss how these data can be helpful to advance our understanding of the topic under study?

-Is public health relevance addressed?

Reviewer #1: The discussion was well-constructed and clearly described how these data enhance understanding of male aegypti biology and mating. Public health relevance would be too extrapolative.

Reviewer #2: (No Response)

**Editorial and Data Presentation Modifications?**

Reviewer #1: Although generally well-prepared, I thought the paper will need some revision to improve clarity. Edits were made directly on the attached manuscript using tracked changes. The authors love to use abbreviations and these should be carefully presented so that the reader does not get lost in the 'alphabet soup'.

Reviewer #2: Recommend making data and code available in a permanent digital repository (e.g., Figshare or Scientific Data) or as supplementary files with the manuscript.

**Summary and General Comments**

Reviewer #1: The authors are to be commended for using a novel marking system to track male movement and mating performance. As they point out, few studies focus on the males although most of the genetic methods focus on male releases. Minor comments have been entered directly on the attached file using tracked changes.

Reviewer #2: My comments are included in the attached PDF. Overall, I found this paper by Trewin et al. to be a very well-written, methodologically sound, and scientifically valuable contribution to the literature on Ae. aegypti movement. The information on spatial patterns of insemination by marked males is especially valuable given the growing interest in SIT/IIT approaches.

The biggest limitation of the paper as written is that it did not acknowledge two key recent papers on movement of Ae. aegypti females and males, which are indicated in my comments (Marcantonio et al. 2019 and Juarez et al. 2020). These papers trapped over broader spatial ranges than most earlier studies and utilized more rigorous modeling approaches (GLMMs and dispersal kernels) very similar to those used in this paper. This resulted in overstatement of the novelty of this study's findings, and the manuscript would benefit from comparing and contrasting the North Queensland results to those of the other studies.

The use of Github and an institutional repository for the modeling code and data from this study are not ideal because there is no guarantee that they will persist into the future. These resources should be preserved as supplementary materials on the PNTDs website or in another permanent digital repository with a DOI.

PLOS authors have the option to publish the peer review history of their article (what does this mean?). If published, this will include your full peer review and any attached files.

Reviewer #1: No

Reviewer #2: No
---

## [Decision Letter · Decision Letter 1]

9 Mar 2021

Dear Dr Trewin,

Thank you very much for submitting your manuscript "Mark-release-recapture of male Aedes aegypti (Diptera: Culicidae): use of rhodamine B to estimate movement, mating and population parameters in preparation for an incompatible male program." for consideration at PLOS Neglected Tropical Diseases. As with all papers reviewed by the journal, your manuscript was reviewed by members of the editorial board and by several independent reviewers. The reviewers appreciated the attention to an important topic. Based on the reviews, we are likely to accept this manuscript for publication, providing that you modify the manuscript according to the review recommendations. 

Sincerely,

Roberto Barrera, Ph.D.

Associate Editor

Eric Dumonteil

Deputy Editor

Reviewer's Responses to Questions

**Key Review Criteria Required for Acceptance?**

**Methods**

-Are the objectives of the study clearly articulated with a clear testable hypothesis stated?

-Is the study design appropriate to address the stated objectives?

-Is the population clearly described and appropriate for the hypothesis being tested?

-Is the sample size sufficient to ensure adequate power to address the hypothesis being tested?

-Were correct statistical analysis used to support conclusions?

-Are there concerns about ethical or regulatory requirements being met?

Reviewer #1: Methods were well done. 

LPI assumptions were not realistic. Assuming a closed population would seem simplistic, when there is a very high recruitment rate from pupal emergence [evidence: low insemination rate among females] and high death rate [low ALE measured by the authors] as well as high vagility of the released mosquitoes. The authors addressed this in the revision by limiting the population estimates to distinct periods post release. This modification would seem adequate for their purposes, but seemingly resulted in an overestimation of population size.

**Results**

-Does the analysis presented match the analysis plan?

-Are the results clearly and completely presented?

-Are the figures (Tables, Images) of sufficient quality for clarity?

Reviewer #1: There appeared to be some discordance between the Figure captions within the text and the numbering of the Figures on the .pdf file. This should be re-examined by the authors.

**Conclusions**

-Are the conclusions supported by the data presented?

-Are the limitations of analysis clearly described?

-Do the authors discuss how these data can be helpful to advance our understanding of the topic under study?

-Is public health relevance addressed?

Reviewer #1: Well done here.

**Editorial and Data Presentation Modifications?**

Reviewer #1: Minor edits to improve clarity were made directly on the attached file using tracked changes. These were minor and can by addressed easily by the authors.

**Summary and General Comments**

Reviewer #1: The manuscript seems about ready to go, after the minor comments on the text were addressed as needed by the authors.

PLOS authors have the option to publish the peer review history of their article (what does this mean?). If published, this will include your full peer review and any attached files.

Reviewer #1: No

Figure Files:

Data Requirements:

Reproducibility:

References

---

## [Editor Report · Decision Letter 2]

1 Apr 2021

Dear Dr Trewin,

We are pleased to inform you that your manuscript 'Mark-release-recapture of male Aedes aegypti (Diptera: Culicidae): use of rhodamine B to estimate movement, mating and population parameters in preparation for an incompatible male program.' has been provisionally accepted for publication in PLOS Neglected Tropical Diseases.

Best regards,

Roberto Barrera, Ph.D.

Associate Editor

Eric Dumonteil

Deputy Editor

---

## [Editor Report · Acceptance letter]

20 May 2021

Dear Dr Trewin,

We are delighted to inform you that your manuscript, "Mark-release-recapture of male *Aedes aegypti* (Diptera: Culicidae): use of rhodamine B to estimate movement, mating and population parameters in preparation for an incompatible male program.," has been formally accepted for publication in PLOS Neglected Tropical Diseases.

Best regards,

Shaden Kamhawi

co-Editor-in-Chief

Paul Brindley

co-Editor-in-Chief
